# Hydrodynamic Characteristics and Conformational Parameters of Ferrocene-Terpyridine-Based Polymers

**DOI:** 10.3390/polym14091776

**Published:** 2022-04-27

**Authors:** Alexander S. Gubarev, Alexey A. Lezov, Nina G. Mikusheva, Igor Perevyazko, Anna S. Senchukova, Alexandra A. Lezova, Anna N. Podsevalnikova, Vyacheslav B. Rogozhin, Marcel Enke, Andreas Winter, Ulrich S. Schubert, Nikolai V. Tsvetkov

**Affiliations:** 1Department of Molecular Biophysics and Polymer Physics, St. Petersburg University, Universitetskaya Nab. 7/9, 199034 Saint-Petersburg, Russia; a.gubarev@spbu.ru (A.S.G.); a.a.lezov@spbu.ru (A.A.L.); n.mikusheva@spbu.ru (N.G.M.); i.perevyazko@spbu.ru (I.P.); grease_91@mail.ru (A.S.S.); a.lezova@spbu.ru (A.A.L.); a.podsevalnikova@spbu.ru (A.N.P.); v.rogozhin@spbu.ru (V.B.R.); 2Laboratory of Organic and Macromolecular Chemistry (IOMC), Friedrich Schiller University Jena, Humboldtstr. 10, 07743 Jena, Germany; marcel.enke@uni-jena.de (M.E.); andreas.winter@uni-jena.de (A.W.); 3Jena Center for Soft Matter (JCSM), Friedrich Schiller University Jena, Philosophenweg 7, 07743 Jena, Germany

**Keywords:** metallopolymers, light scattering, analytical ultracentrifugation, polymer chain conformation, equilibrium rigidity, complexation

## Abstract

Nowadays, the study of metallopolymers is one of the fastest growing areas of polymer science. Metallopolymers have great potential for application in multiple technological and various biomedical processes. The macromolecules with the possibility of varying the number and type of metal ions along the entire length of the polymer chain are of particular interest. In this regard, this study presents results on two successfully synthesized homopolymers, random and block copolymers based on PMMA, containing ferrocene and terpyridine moieties in the side chain. Different architectures of copolymers may attribute interesting properties when creating complexes with various metal ions. A detailed hydrodynamic study of these structures was carried out, the consistency of hydrodynamic data was established using the concept of a hydrodynamic invariant, the absolute values of the molar masses of the studied objects were calculated, and the conformational parameters of macromolecules were determined. Using the Fixman–Stockmayer theory, the equilibrium rigidities of the studied systems were calculated and the relationship between the chemical structure and conformational characteristics was established. The studied copolymers can be attributed to the class of flexible-chain macromolecules. An increase in the equilibrium rigidity value with an increase of the side chain, which is characteristic of comb-shaped polymers, was determined.

## 1. Introduction

Polymers containing metal coordination centers find application in, for instance, catalytic processes [1,2,3], electronics [4,5,6], nanoscience [7], biomedical, and healthcare applications [8,9]. Such materials combine the processing advantages of polymers with the functionality provided by the presence of metal centers. They can serve as smart coatings [10,11], self-healing materials [12,13,14,15,16,17,18,19,20], energy storage [21,22,23] and memory devices [24,25], biodegradable packaging [26], and more [27,28,29,30,31].

Until the end of the last century, the metallopolymer area was restrained by synthesis difficulties. Specifically, the conventional polymerization protocols that work well for organic monomers are often incompatible with metal-containing systems, and due to insufficient chain growth and undesirable side reactions, metallopolymers were typically obtained with poor yields and low molar masses. However, this shortcoming could be overcome by the development of the various controlled polymerization techniques, which tolerate the presence of metal centers. Thus, metallopolymers can now be prepared in reasonable amounts, and a broad range of applications have started to emerge [32,33,34]. Nowadays, a wide range of metal-containing polymers can be prepared with a good control over the molar mass and with narrow dispersity [35,36].

Metallopolymers can contain a variety of metal centers, from main group metals such as Sn and Pb, to transition metals such as Fe, Ir, and Pt, through to lanthanides such as Eu. The metals can be located either in the polymer main chain [16,37,38] or in the side chains of a comb-like polymer. The side chains, in turn, can be comprised of metallocene-containing units [4,7,8,9], or contain a complex formed from chelating units [3,39].

A promising approach in this field is the conjunction of covalently linked polymer systems with non-covalent (e.g., electrostatic) interactions to establish new polymeric systems. In this case, interesting properties of the material can occur due to differences in the chemical stability of the bonds.

A new one-step synthesis of polymeric systems combining ferrocene moieties and Pt^2+^ complexes in one polymer was reported in 2008 [40]. In a few subsequent reports, different types of metal-containing ligands were also combined [41,42,43,44].

Research in this area has mainly been focused on optimizing the preparation, enabling a reliable characterization, and tuning the performance of the materials with respect to the fields of interest. So far, the physical characteristics of metal-containing polymers in solution have hardly been looked at.

An understanding of how the distribution of charged metal sites along a polymer chain affects the conformation and/or aggregation in solution could be of central importance to optimize the structure and, thereby, to improve the required properties (e.g., intra- vs. inter-chain energy-transfer processes, accessibility of catalytically or pharmaceutically active centers, etc.).

Though improved characterization techniques made structural analysis easier [45,46,47,48,49,50], the precise characterization of metal-containing polymers remains a challenging task. At this point, it is crucially important to pay attention on the correlation between molecular characteristics received by independent methods and self-consistency of the obtained data. Additional parameters, such as the role of the counter ions or the solubility in various solvents, have to be considered. Thus, the accurate characterization of polymers with ferrocene-containing units can be considered as a separate task necessarily preceding interpretation of data on more complicated structures.

The combination of ligands containing different metal ions can provide outstanding material properties. Since pyridine and its derivatives are well-known for being able to form coordination complexes with metal-ion centers [51], the combination of ferrocene and terpyridine units within one chain is a promising approach for the formation of complexes combining Fe and such ions as Pt^2+^ (or Eu^3+^, Pd^2+^, etc. [52]). In this respect, the preparation of ferrum–platinum nanoparticles, which possess remarkable magnetic properties, represents a notable application [40,53]. It has been demonstrated that such bimetallic nanoparticles can be derived from metallopolymers, which contain Fe and Pt centers, by pyrolysis. Regarding the formation of well-defined nanoparticles, various parameters have to be considered, such as distribution of the metal centers along the polymer chains, conformation of the polymer chains, and their self-assembly behavior [40,54]. Talking about the effect of the presence of metal ions on conformational changes, the comparison of Fe-Pt-containing polymers with Fe-containing chains of the same composition (except for the presence of Pt) is of particular interest in the perspective of the proposed study.

The conformational characteristics of comb-like polymers largely depend on the volume and density of the distribution of the side chains [55]. Thus, it is crucial to correlate changes in the composition and structure of the side chains to the conformational properties of the whole macromolecule. Within the framework of this discourse, the authors compare the properties of the metallopolymers studied herein and the polymer analogous structure—poly(methyl methacrylate) (PMMA)—being the basis for the Fc/tpy metallopolymers (Fc: ferrocene; tpy: 2,2′:6′,2″-terpyridine).

This article is organized as follows: Firstly, the synthesis and structural characterization of four initial polymer structures (i.e., the homopolymers containing either Fc or tpy moieties as well as statistic and block copolymers, which contain both entities) are presented in detail. Secondly, the complete hydrodynamic analysis of these polymers is performed in order to obtain the absolute values of molar masses of the studied polymers and establish a self-consistency check of hydrodynamic data within the concept of the hydrodynamic invariant. Finally, the conformational parameters of the polymers studied herein are evaluated and compared to the metallopolymer carrier backbone structure (PMMA).

## 2. Materials and Methods

### 2.1. Materials

All chemicals, reagents, and solvents, which were required for the synthesis of monomers and polymers, were obtained from Sigma-Aldrich Chemie GmbH (Taufkirchen, Germany) or TCI (Eschborn, Germany) and used as received, unless stated otherwise. Toluene was purified by distillation from Na metal. Azobisisobutyronitrile (AIBN) was recrystallized from ethanol prior to its use.

For performing solubility checks and obtaining preliminary data on polymers, toluene and dimethylformamide (DMF) were used. The solvents were obtained from Vekton (St. Petersburg, Russia) and used as received. For the majority of hydrodynamic experiments, tetrahydrofuran (THF) (Vekton, St. Petersburg, Russia) was used. It was purified by distillation under an atmosphere of dry argon in the presence of sodium hydroxide and hydroquinone. While ascertaining the data on partial specific volume, the deuterated tetrahydrofuran (THF-d8 > 99.5% atom D) was used, which was purchased from Sigma-Aldrich. The parameters (density, $\rho_{0}$, and dynamic viscosity, $\eta_{0}$) of all the solvents were obtained experimentally and the obtained values are presented in Appendix A.

### 2.2. Polymer Synthesis

The protocols for synthesizing the monomers of ferrocene and terpyridine were described elsewhere [56,57]. The synthesis of the polymers via reversible addition fragmentation chain-transfer polymerization (RAFT) was performed in oven-dried glassware. In general, a microwave vial (5 mL) was charged with the respective monomers and toluene stock solutions, which contained the initiator (AIBN) and the chain-transfer agent (2-cyano-2-propyl dithiobenzoate, CPDB), were added. The resulting solution was cautiously degassed by purging with a N_2_ stream for at least 30 min. Subsequently, the sealed vial was heated to 70 °C for 12 h. The crude polymer was precipitated by dropping the reaction mixture into ice-cold MeOH (20 mL). The monomers were removed by dialysis in THF using a size-exclusion membrane (MWCO 3300 Da). The obtained polymers were structurally characterized by proton nuclear magnetic resonance spectroscopy (^1^H-NMR) and size-exclusion chromatography (SEC). Following this general protocol, the Fc- and tpy-containing homopolymers as well as the statistic copolymer were synthesized (Figure 1). The block copolymer (Fc)-*bl*-(tpy) was prepared in a similar fashion, but using *poly*(tpy) as the macromolecular chain-transfer agent. Further details on the polymer synthesis are provided in Appendix A. The ^1^H-NMR spectra of all synthesized samples are presented in Appendix A, and the data of SEC analyses are summarized in Table 1 and Figure 2.

### 2.3. Methods

**NMR.** ^1^H NMR spectra were recorded on a Bruker (Karlsruhe, Germany) AVANCE I 300 MHz instrument in CDCl_3_ or CD_2_Cl_2_ (Euriso-Top GmbH, Saarbrücken, Germany) at 25 °C. Chemical shifts are reported in ppm and are referenced using the residual solvent signal.

**SEC.** SEC measurements were performed using a Shimadzu (Duisburg, Germany) system comprising of a CBM-20A system controller (system controller), a DGU-14A degasser, a LC-10AD VP pump, a SIL-10AD VP auto sampler, a refractive-index (RID-10A) detector, and a PSS GRAM guard/1000/30 Å column. The data were recorded using CHCl_3_/isopropanol/NEt_3_ (94:2:4 volume ratio) as the eluent, at 40 °C and at a flow rate of 1 mL/min. The obtained data were analyzed by applying a linear poly(methyl methacrylate) or polystyrene calibration (PSS GmbH).

**Viscometry**. The rolling-ball micro-viscometer Lovis 2000 M (Anton Paar, Graz, Austria) was used to determine the values of intrinsic viscosity [η] via Huggins and Kraemer plots [58,59]. Whenever it was not possible to resolve concentration dependences, the Solomon–Ciuta equation was used for estimation of [η] by performing viscosity measurements at a single solution concentration, c [60]. The values of relative viscosity, $\eta_{r}=\eta /\eta_{0}$, are equal to the $t/t_{0}$ ratio, where and $t_{0}$ are the solution and solvent rolling times, correspondingly. According to Huggins, the concentration dependence of specific viscosity divided by solution concentration, $\eta_{sp}/c=\left(\eta_{r}-1\right)/c$, can be approximated by the following equation: $\left[\eta \right]+k^{\prime }{\left[\eta \right]}^{2}c$. In the case of the Kraemer plot, the dependence of $ln\eta_{r}/c$ vs. solution concentration equals $\left[\eta \right]+k^{\prime \prime }{\left[\eta \right]}^{2}c$. The intercept of both of the specified plots is intrinsic viscosity [η], and Huggins $k^{\prime }$ and Kraemer $k^{\prime \prime }$ constants depend on the solvent quality and could be evaluated from the slopes of corresponding graphs. The performed measurements were accomplished at a low concentration range, at $t/t_{0}<2.5$, and approximated by straight lines (Figure 3). All of the experiments were performed at a constant temperature of 20 °C, the capillary inner diameter was 1.59 mm, the gold-coated steel ball diameter was 1.50 mm, and the capillary inclination angle was 40°.

**Analytical ultracentrifugation (AUC).** The velocity sedimentation experiments were performed using the analytical ultracentrifuge “ProteomeLab XL-I Protein Characterization System” (Beckman Coulter, Brea, CA, USA) at 20 °C, and the rotor speed varied from 30,000 to 40,000 rpm depending on the sample. Sedimentation was observed using mainly the Rayleigh interference optical system equipped with a red laser (*λ* = 655 nm) as a light source and double-sector cells with aluminum centerpieces with an optical path length of 12 mm. The sample and the reference sectors were loaded with 0.44 mL of studied solution and a solvent, respectively. The centrifuge chamber with a loaded rotor and interferometer was vacuumed and thermo-stabilized for at least 60 to 90 min before the run.

The experimental data were processed with the use of the Sedfit program [61], allowing to obtain sedimentation coefficients’ distribution (Figure 4) using the Provencher regularization procedure [62,63]. The sedimentation coefficient, s, and the frictional ratio, $\left(f/f_{sph}\right)$, were determined. The frictional ratio values allow to perform an estimation of diffusion coefficients, but the self-consistency of acquired data must be established [64,65,66]. Both of the characteristics, s and $\left(f/f_{sph}\right)$, should be extrapolated to infinite dilution. Since the hydrodynamic investigations are usually performed in very dilute solutions, the linear approximations $s^{-1}=s_{0}^{-1}\left(1+k_{s}c\right)$ and $f/f_{sph}={\left(f/f_{sph}\right)}_{0}\left(1+k_{f}c\right)$ (where $k_{s}$ is the Gralen coefficient, *f* is the translational friction coefficient, and $f_{sph}$ is the translational friction coefficient of the equivalent sphere) can be used. In this way, the hydrodynamic parameters $s_{0}$ and ${\left(f/f_{sph}\right)}_{0}$, characterizing a macromolecule at the infinite dilution limit, can be determined. The *D* value may be extracted in some cases from the frictional ratio calculated by the Sedfit program: $D_{0sf}=\frac{k_{B}T{\left(1-\bar{\upsilon }\rho_{0}\right)}^{1/2}}{\eta_{0}^{3/2}9\pi \sqrt{2}{\left({\left(f/f_{sph}\right)}_{0}\right)}^{3/2}{\left(s_{0}\bar{\upsilon }\right)}^{1/2}}$ (where $k_{B}$, *T*, and $\bar{\upsilon }$ are the Boltzmann constant, the absolute temperature, and the partial specific volume, correspondingly) [61].

**Partial Specific Volume**. The density measurements were carried out in THF and DMF solutions at 20 °C, using the densitometer DMA 5000 M (Anton Paar, Graz, Austria) according to the procedure described elsewhere [67]. This procedure involves obtaining solution density, ρ (or difference between solution and solvent density: $\Delta \rho =\left(\rho -\rho_{0}\right)$, where $\rho_{0}$ is the solvent density), data vs. its concentrations. The corresponding dependences are plotted in Figure 3. The slopes of the extrapolated linear dependences correspond to the value of the buoyancy factor, ∂Δρ/∂c. The value of the partial specific volume was calculated as $\bar{\upsilon }=\left(1-\partial \Delta \rho /\partial c\right)/\rho_{0}$.

**Dynamic light scattering (DLS)**. Autocorrelation functions of dynamic light scattering were acquired on the “PhotoCor Complex” apparatus (Photocor Instruments Inc., Moscow, Russia). Analysis of the functions was performed in DynaLS software, which allows to obtain a distribution of relaxation times, τ(ρ(τ)), from regularized inverse Laplace transform. Diffusion coefficients, τ(ρ(τ)), of the studied macromolecules at finite concentration were determined from linear dependences $1/\tau =Dq^{2}$, where q=4πn/λsin(ϑ/2) is a scattering vector [68,69,70]. The values of diffusion coefficients of individual macromolecules, $D_{0}$, were determined from extrapolation of concentration dependences to infinite dilution. All DLS studies were performed at a constant temperature of 20 °C, the laser light wavelength λ = 654 nm, and the scattering angle (ϑ) was varied from 30° to 130°. The refractive index of the solvent *n* was determined by the DM40 (Mettler Toledo, Greifensee, Switzerland) refractometer. Hydrodynamic radius, $R_{h}$, was calculated using the Stokes–Einstein equation [55]:(1)$$R_{h}=\frac{k_{B}T}{6\pi \eta_{0}D}.$$

## 3. Results

The results of the viscosity study are presented in Figure 3a. All the obtained dependences are well-fitted by straight lines and their intersections with the ordinate axis provide the values of the intrinsic viscosity [η] for all studied polymers. The obtained values of intrinsic viscosities are relatively low, meaning that the studied polymers belong to the moderate range of molar masses and/or the structure of the polymer coils is compact and of high density (Table 2). Both of these assumptions correlate well with the aforementioned SEC data (Table 1) and with the fact that most of the studied polymers bear metal ions.

The reliable determination of the partial specific volume, $\bar{\upsilon }$, is important for calculation of the absolute molar mass, $M_{sD}$:(2)$$M_{sD}=\frac{s_{0}RT}{D_{0}\left(1-\bar{\upsilon }\rho_{0}\right)}=\frac{\left[s\right]}{\left[D\right]}R,$$
where R represents the universal gas constant, and $\left[s\right]\equiv \frac{s_{0}\eta_{0}}{\left(1-\bar{\upsilon }\rho_{0}\right)}$ and $\left[D\right]\equiv \frac{D_{0}\eta_{0}}{T}$ are the intrinsic values of the velocity sedimentation coefficient [s] and the translational diffusion coefficient [D], respectively. Usually, the buoyancy factor is determined by measuring the dependence of solution density vs. concentration (Figure 3b). However, such approach might obtain a false result due to limitations of the experimental technique dealing with an additive characteristic value, which represents the value of the partial specific volume [71,72].

An alternative technique (known as the density variation approach) requires isotopically different solvents, which differ in basic solvent parameters (i.e., density and dynamic viscosity), and at the same time the conformational status of macromolecules is not affected [73]. Then, the comparison of sedimentation coefficients, obtained in THF and THF-d8, should afford the genuine value of the partial specific volume:(3)$${\bar{\upsilon }}_{dva}=\frac{s_{THF}\eta_{0THF}-s_{THFd8}\eta_{0THFd8}}{s_{THF}\eta_{0THF}\rho_{0THFd8}-s_{THFd8}\eta_{0THFd8}\rho_{0THF}}$$

This approach also has some limits of application. As was shown elsewhere [74], the solutions of salts in regular and deuterated solvent reveal different structures. The accuracy of determining the partial specific volume in this manner is also negatively affected by the polydispersity of the studied polymers. Nevertheless, both of the above methods were implemented, and fortunately the obtained results were found to be consistent (Table 2).

In further calculations, the averaged values, $\langle \bar{\upsilon }\rangle$, are used. The obtained values for the partial specific volume were compared to known literature data. Thus, the value of 0.62 cm^3^/g obtained for *poly*(Fc) is well-correlated with (0.62 ± 0.01) cm^3^/g determined for a coordination array on the basis of Co(II) ions and the ditopic 4,6-*bis*(6-(2,2′-bipyridil))pyrimidine [75] and, on the other hand, 0.75 cm^3^/g resolved for *poly*(tpy) is approaching to the partial specific volume range (0.78 to 0.87) cm^3^/g known for PMMA and DMAEMA [50,71].

In the first stage of preliminary work with the systems, the solubility studies were performed with velocity sedimentation experiments using the AUC. As a test system, the *poly*(Fc) was chosen and dissolved in toluene, THF, and DMF. The results are presented in Figure 4a.

While *poly*(Fc) afforded nearly identical distributions of the sedimentation coefficients in toluene and THF (Figure 4a), the solubility issues were found in toluene solutions. Some parts of the sample were not dissolved—independent of time, temperature, and other applied factors. On the other hand, the data resolved for the DMF solutions revealed quite prolonged distribution fractions positioned in high molar mass regions. This fact might indicate partial macromolecule aggregation in this solvent. Based on the acquired data, THF was found to be the best candidate of those tested, and thus the majority of the subsequent hydrodynamic experiments were conducted in THF solutions.

The results shown in Figure 4b agree with the SEC data (Table 1). The characteristic values of the sedimentation coefficients are proportional to the molar mass values (Equation (2)). It should be noted that the distributions for *poly*(Fc), *poly*(tpy), and ((Fc)-*co*-(tpy)) were found nearly in the same region of sedimentation coefficients, whereas the distribution for *poly*(Fc) was the only one to present one major peak—the other polymers had several prominent peaks attributed to their higher dispersity. The concentration dependency of sedimentation coefficients has been resolved within at least a 3-fold concentration range relative to the lowest concentration value. From these experiments, the sedimentation coefficients at the infinite dilution limit were determined (Appendix A).

The initial DLS data (inverse relaxation times, 1/τ, vs. scattered vectors squared, $q^{2}$) are presented in Appendix A. The scattered light-intensity distributions over the hydrodynamic radii of the studied systems were characterized by several peaks at all studied concentrations (Appendix A). It is possible to calculate the weight concentration of the solution components in accordance with the expression $w\sim I/R_{h}^{3}$ (here, w represents the mass fraction of the system component, I is the scattering intensity, and *R_h_* is the linear size of the scattering centers) (Figure 5). The performed analysis demonstrates that the main fraction of the studied samples in the solution is represented by particles with a hydrodynamic radius, *R_h_*, within the range from 1 to 10 nm. Apparently, this peak corresponds to the major diffusion component of the studied polymers and, consequently, the obtained hydrodynamic radii, *R_h_*, values correspond to the molecular sizes of the studied (co)polymers. The diffusion coefficient, $D_{0}$, at the infinite dilution limit was obtained by extrapolating to zero concentration, and the *R_h_* values obtained in this way were recalculated to diffusion coefficients according to Equation (1) at all studied concentrations (Appendix A). The mass fraction of larger particles did not exceed a few percent.

At this point, the complete set of hydrodynamic parameters is obtained and the mutual agreement of hydrodynamic characteristics should be established. This goal is usually achieved within the concept of the hydrodynamic invariant, $A_{0}$ [76,77]:(4)$$A_{0}={\left(R\left[s\right]{\left[D\right]}^{2}\left[\eta \right]\right)}^{\frac{1}{3}}$$

To accomplish this check, the sedimentation, *s*_0_, and diffusion, *D*_0_, coefficients determined for the infinite dilution limit have to be expressed as intrinsic values ([*s*] and [*D*], correspondingly). These values of hydrodynamic parameters are independent of common solvent parameters and correlated with the molar mass and hydrodynamic size of a macromolecule.

The average values of the hydrodynamic invariant, $A_{0}$, were calculated, which were within the limits of insignificant statistical deviations determined by the sensitivity of the applied experimental techniques (Table 3). It means that a satisfactory correlation between the molecular characteristics (i.e., [η], [s], [D]) was obtained, which in turn makes it possible to interpret the experimental data. The average values of $A_{0}$ and $A_{0sf}$ (which operates with estimated diffusion coefficients based on frictional ratio values determined with Sedfit analysis) for all studied systems were found to be $\left(2.8\pm 0.1\right)\times 10^{-10}$ and $\left(3.4\pm 0.1\right)\times 10^{-10}\;{g\;\mathrm{cm}\;s}^{-2}{\;K}^{-1}{\;\mathrm{mol}}^{-1/3}$, respectively. The hydrodynamic invariant calculated for ((Fc)-*bl*-(tpy)) based on the frictional ratio value (for polydisperse systems, this value can be determined incorrectly), as well as the data obtained from DMF solution, were not included in averaging. The higher values of molar mass obtained in DMF solutions also indicate the macromolecular aggregation.

The calculated values of the hydrodynamic invariant belong to the range known for compact, non-percolated macromolecules [76,78]. Further, the absolute values of the molar masses can be determined using the Svedberg equation (Equation (2)). Although $M_{sD}$ and $M_{sf}$ values were found in satisfactory agreement for the homopolymers in THF and toluene solutions, it should be noted that the $M_{sD}$ values were calculated based on the data from two independent experimental techniques and should be treated as more equitable results. On the other hand, the evaluation of $M_{sf}$ values can be imprecise when dealing with polydisperse samples, which is exactly the case for the block copolymer solutions in THF.

## 4. Discussion

PMMA represents a well-known polymer and its conformational characteristics have been obtained and analyzed in sufficient detail [50,79,80]. It is known that the presence of the large side chains in the monomers composing a linear macromolecule leads to an increase in the equilibrium rigidity [55]. The studied polymers and copolymers with ferrocene and terpyridine side substituents are capable to exhibit such behavior.

Usually, the main conformational parameter, the value of equilibrium rigidity, is determined based on the analysis of one of the hydrodynamic characteristics (i.e., intrinsic viscosity or diffusion/sedimentation coefficient) of a polymer homologous series vs. molar mass values [81,82,83]. The comparative analysis of the equilibrium rigidity of the studied systems was performed on the basis of the Fixman–Stockmayer theory [84]. Within the framework of this model, a relation was obtained for the value of intrinsic viscosity, which makes it possible to determine the value of equilibrium rigidity based on the following extrapolation:(5)$$\frac{\left[\eta \right]}{M^{1/2}}\frac{M_{L}^{3/2}}{\Phi_{0}}=A^{3/2}+0.51BM^{1/2}M_{L}^{3/2},$$
where $M_{L}=M_{0}/\lambda$ is the molar mass per unit chain length, λ is the projection of a monomer unit to the direction of the fully extended polymer chain, $M_{0}$ is the molar mass of a monomer unit, $\Phi_{0}=2.87\times 10^{23}\;{\mathrm{mol}}^{-1}$ is the Flory hydrodynamic parameter [85], *A* is the equilibrium rigidity, and B is the model parameter, characterizing the thermodynamic quality of a solvent. In order to solve these equations, the $M_{L}$ value must be known. The $M_{L}$ values were obtained based on the $M_{0}$ values calculated for all of the polymers/copolymers and the known $\lambda =2.52\times 10^{-8}\;\mathrm{cm}$ value characterizing the alkyl chain (Table 4).

The results obtained by extrapolation (Equation (5)) of already known data for PMMA and its comparison with the studied samples are presented in Figure 6. The slope of this dependence characterizes the thermodynamic quality of the solvent. It is not possible to draw such a conclusion by studying a single sample of individual composition. This is the aim to be achieved in further studies.

In our case, in the absence of polymer homologous series of the studied (co)polymer samples, the aforementioned method allows to estimate only the top limit, $A_{app}$, for the values of equilibrium rigidity: $\left[\eta \right]M_{L}^{\frac{3}{2}}/\left(M^{\frac{1}{2}}\Phi_{0}\right)\approx A_{app}{}^{\frac{3}{2}}$. In the occasion of possible circumstances, when the model coefficient B=0 (which corresponds to thermodynamically ideal conditions), the top limit of the values of equilibrium rigidity, $A_{app}$, will coincide with *A*. The bottom limit value of equilibrium rigidity can be correlated with the equilibrium rigidity determined for PMMA (1.4 nm).

The data on the top limit $A_{app}$ for the values of equilibrium rigidity are summarized in Table 4. It was found that the highest $A_{app}$ value was obtained for *poly*(tpy), which bears bulky tpy side groups (5.4 nm), while the lowest one corresponds to the *poly*(Fc) (3.5 nm). Such an increase in $A_{app}$ in the studied systems in comparison with PMMA can most likely be associated with the presence of a large side substituent of the ferrocene and terpyridine polymers, and a similar effect was observed for comb-like polymers [55]. The terpyridine side chain was bulkier than the ferrocene one (see Figure 1), which was expressed by an almost four-fold difference between $A_{\mathrm{PMMA}}$ and $A_{app}$. For *poly*(Fc), the $A_{app}$ was about three times higher than *A* for PMMA.

For block copolymer (Fc)-*bl*-(tpy), the $A_{app}/A_{\mathrm{PMMA}}$ ratio was found to be slightly lower compared to the copolymer with a statistical distribution of monomers, (Fc)-*co*-(tpy). Usually, the estimation of the values of the equilibrium rigidity for copolymers can be accomplished independently by assuming the additivity of polymer chain flexibility. This approach operates with the sum of reciprocal values of equilibrium rigidity of the respective homopolymers, which constitute the chemical composition of the copolymer: $A_{app}{}^{-1}=\frac{n}{\left(m+n\right)}A_{app\_poly\left(\mathrm{Fc}\right)}^{-1}+\frac{m}{\left(m+n\right)}A_{app\_poly\left(\mathrm{tpy}\right)}^{-1}$ (where and m are coefficients attributed to composition). The above-described approach can be used for an assessment of for the statistic and block copolymers (their composition is specified in Figure 1). It resulted in the values of $A_{app\left(co\right)}=4.2\;\mathrm{nm}$ for (Fc)-*co*-(tpy) and $A_{app\left(bl\right)}=4.9\;\mathrm{nm}$ for (Fc)-*bl*-(tpy), correspondingly. As can be seen, these values are within the range of $A_{app}$ values determined via the Fixman–Stockmayer theory for homopolymers, although there was no satisfactory correlation with the $A_{app}$ determined for copolymers. It can be assumed that the obtained result reflects the fact that steric hindrances of rotation for the studied statistical copolymer are not well-described within the flexibility-additivity approach or demonstrates the necessity to take volume effects into consideration for the studied system.

Finally, another important conformational parameter can be determined for the studied samples: the diameter of a polymer chain (transverse diameter). These data can be estimated based on the values of the partial specific volume (Table 2) using the following equation [86]:(6)$$d=\sqrt{\frac{4M_{0}\bar{\upsilon }}{\pi \lambda N_{A}}},$$
where $N_{A}$ represents Avogadro’s constant. In such a manner, calculated values are presented in Table 4. The values of the polymer chain diameter obtained for homopolymers *poly*(Fc) and *poly*(tpy) were 1.38 and 1.62 nm, respectively. Homopolymer *poly*(tpy) has the bulkiest side chain, which makes the biggest value of the diameter reasonable and expected. Note that it is twice as high as the diameter of the PMMA side chains. The values of the diameters found for copolymers ((Fc)-*co*-(tpy)) and ((Fc)-*bl*-(tpy)) were equal to 1.47 and 1.50, respectively. This fact is also logical as the acquired values are determined by the data range obtained for the homopolymers.

The last column of Table 4 represents the value of the average number of the polymer chain segments, $N=L/A_{app}$, where *L* is the contour length of a polymer chain. For the majority of the studied samples, N is close to 10, except for the ((Fc)-*bl*-(tpy)) copolymer, which has N~37. All of the studied samples can be related to the formed polymer coils, and the obtained values of equilibrium rigidity and transverse diameter are common to comb-like polymers.

## 5. Conclusions

The presented study has reported on the successful synthesis of four systems: the homopolymers containing either Fc or tpy moieties, as well as statistic and block copolymers, which contain both entities. A detailed hydrodynamic study of these structures was carried out, and the consistency of hydrodynamic data was established using the concept of a hydrodynamic invariant. The average value of the hydrodynamic invariant attributed to all studied systems was found to be $\left(3.4\pm 0.1\right)\times 10^{-10}{\;g\;\mathrm{cm}\;s}^{-2}{\;K}^{-1}{\;\mathrm{mol}}^{-1/3}$, which is a characteristic of compact, non-percolated macromolecules. Further, the absolute values of the molar masses of the studied objects were calculated, and the conformational parameters of macromolecules were determined. Using the Fixman–Stockmayer theory, the equilibrium rigidities of the studied systems were calculated, and the relationship between the chemical structure and conformational characteristics was established. The equilibrium rigidity values of the synthesized samples were found in the interval defined by the structures of homopolymers *poly*(Fc) and *poly*(tpy) and constituted 3.5 to 5.4 nm, correspondingly. The equilibrium rigidity of statistic and block copolymers had moderate values entering the above-mentioned interval: 4.6 and 4.3 nm, correspondingly. The conformational parameters of the studied polymers were compared to those of PMMA, which resulted in an almost three- to four-fold difference in equilibrium rigidity values for *poly*(Fc) and *poly*(tpy). The studied copolymers can be attributed to the class of flexible-chain macromolecules. An increase in the equilibrium rigidity values of tpy-containing polymers was associated with an increase of the side chain, which is a characteristic of comb-shaped polymers. It should be noted that the results of the systems studied here became the basis for a consequent study on complexation of *poly*(tpy) and (Fc)-*co*-(tpy) with Eu^3+^ and Pd^2+^ ions [52].

## Figures and Tables

**Figure 1 polymers-14-01776-f001:**
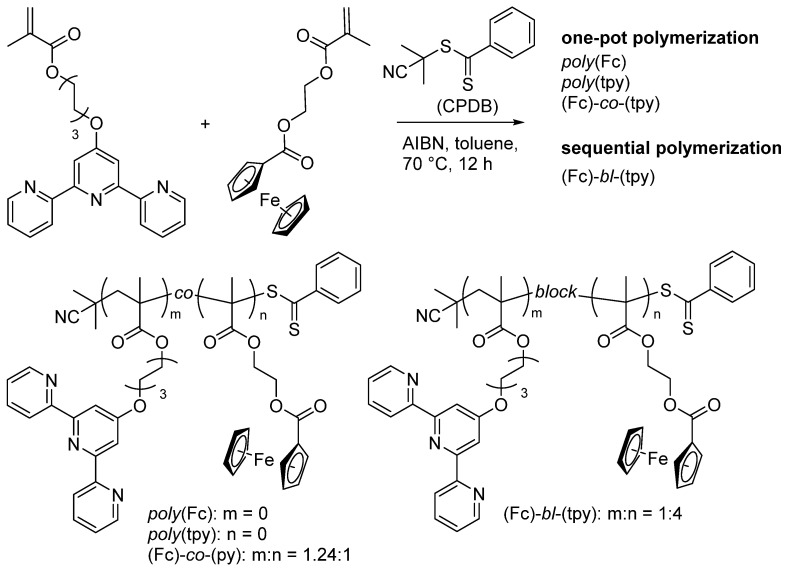
Schematic representation of the synthesis of four Fc- and/or tpy-containing polymers.

**Figure 2 polymers-14-01776-f002:**
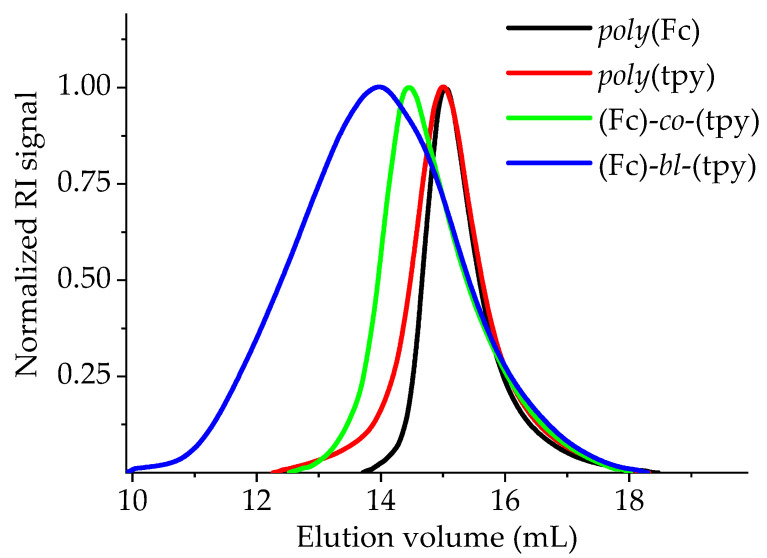
Normalized SEC curves of the four polymers (eluent: CHCl_3_/isopropanol/NEt_3_ in a 94:2:4 volume ratio).

**Figure 3 polymers-14-01776-f003:**
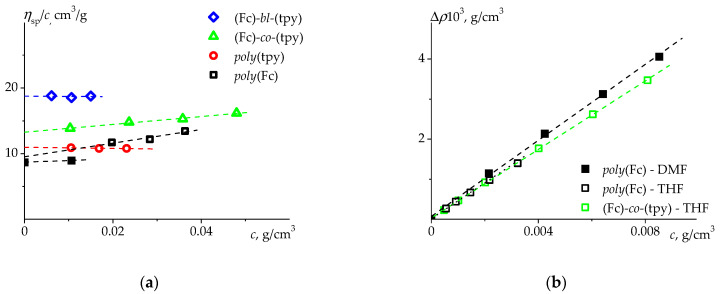
(**a**) The normalized specific viscosity, *η_sp_*/*c*, vs. concentration, *c*, obtained for the studied polymers in THF (open symbols) and DMF (filled symbols) at 20 °C. (**b**) The dependence of Δ*ρ* = *ρ − ρ*_0_ on the polymer concentration, *c*, determined for *poly*(Fc) and (Fc)-*co*-(tpy) at 20 °C.

**Figure 4 polymers-14-01776-f004:**
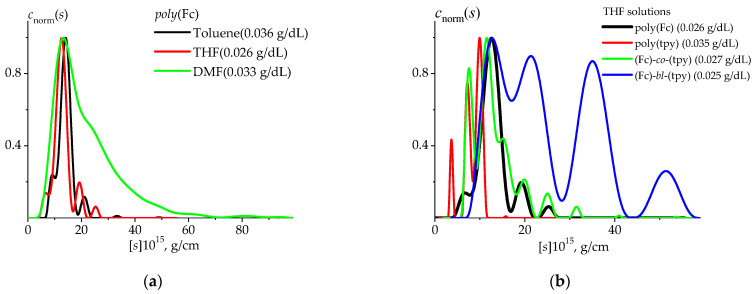
Normalized *c*(*s*) distributions vs. intrinsic sedimentation coefficients [*s*] resolved with Sedfit for the *poly*(Fc) sample in different solvents (**a**) and all of the studied polymers in THF solutions (**b**). The presented distributions correspond to the lowest studied concentration of the systems, and the concentration values are indicated in the figure. All AUC runs were performed at 20 °C.

**Figure 5 polymers-14-01776-f005:**
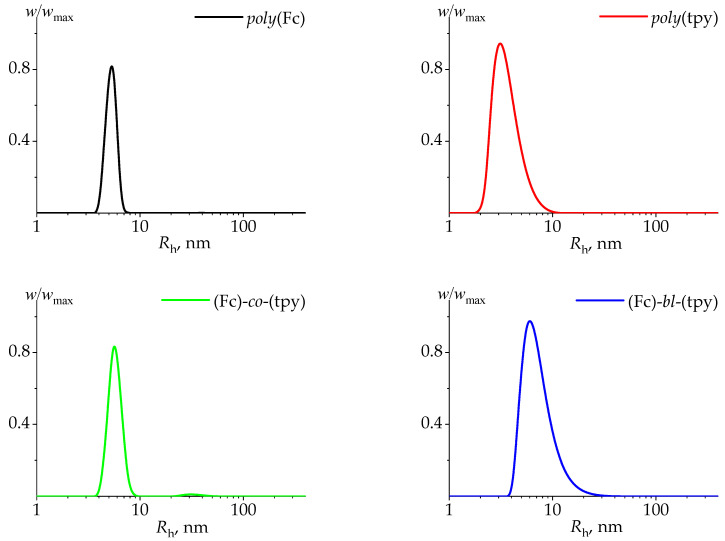
The normalized distributions of weight component concentration, *w*/*w_max_*, over hydrodynamic radii, *R_h_*, obtained with DLS for the studied polymers (*poly*(Fc), *poly*(tpy)) and copolymers ((Fc)-*co*-(tpy), (Fc)-*bl*-(tpy)) in THF solutions. All data were obtained at 20 °C. The diffusion coefficients, *D*_0_, at the infinite dilution limit are presented in Table 3.

**Figure 6 polymers-14-01776-f006:**
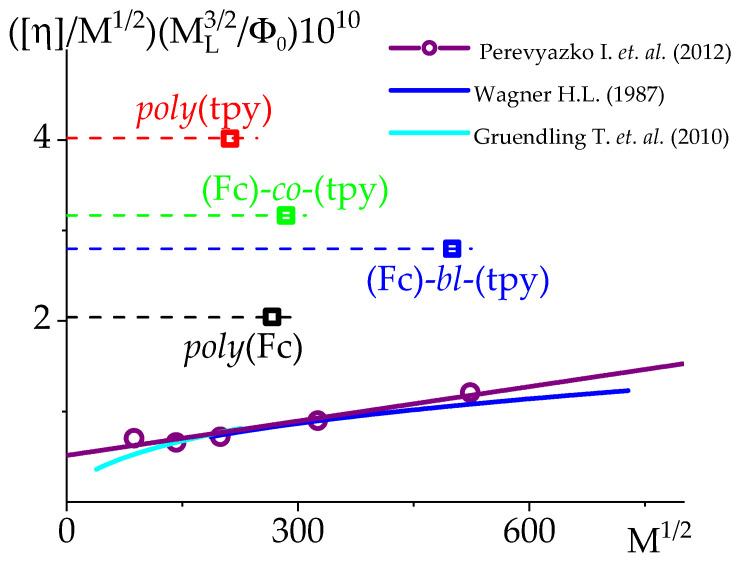
The application of Fixman–Stockmayer theory to the hydrodynamic data based on already published PMMA data [50,79,80] (solid lines and open circles) and its comparison with the studied samples (dashed lines and open squares).

**Table 1 polymers-14-01776-t001:** Summary of the SEC data: number-averaged molar mass (*M*_n_) and molar mass distribution (dispersity, *Đ = M*_W_/*M*_n_, where *M*_W_ is the weight-averaged molar mass) ^1^.

Sample	*M*_n_, g/mol	*Đ*
*poly*(Fc)	23,900	1.20
*poly*(tpy)	24,600	1.41
(Fc)-*co*-(tpy)	29,000	1.44
(Fc)-*bl*-(tpy)	46,000	2.70

^1^ Eluent: CHCl_3_/isopropanol/NEt_3_ in a 94:2:4 volume ratio. *M*_n_ values were estimated using a linear PS calibration.

**Table 2 polymers-14-01776-t002:** The partial specific volume values, obtained with densitometry (${\bar{\upsilon }}_{d}$) in THF and density variation approach (${\bar{\upsilon }}_{dva}$) in THF vs. THF-d8 solutions at 20 °C.

Sample	${\bar{\upsilon }}_{d},$ cm^3^/g	${\bar{\upsilon }}_{dva},$ cm^3^/g	$\langle \bar{\upsilon }\rangle ,$ cm^3^/g
*poly*(Fc)	0.65 0.57 ^1^	0.63	0.62 ± 0.03
*poly*(tpy)	-	0.75	0.75 ± 0.04
(Fc)-*co*-(tpy)	0.65	0.65	0.65 ± 0.02
(Fc)-*bl*-(tpy)	-	0.71	0.71 ± 0.05

^1^ Determined in a DMF solution (Figure 3).

**Table 3 polymers-14-01776-t003:** The hydrodynamic parameters, hydrodynamic invariants, and molar masses of the studied Fc-/tpy-containing (co)polymers defined in the specified solvents at 20 °C.

Sample	Solvent	[*η*], cm^3^/g	*s*_0_, S	(*f*/*f*_sph_)_0_	D_DLS_10^7^, cm^2^/s	*R*_h_, nm	A_0_10^10^/ A_0sf_10^10^	M_sD_10^−3^/ M_sf_10^−3^, g/mol
*poly*(Fc)	THF Toluene DMF	9.2	11.5	1.61	8.75	4.9	**2.8**/3.3	**71**/56
			10.7	1.46	8.59	4.3	**3.1**/3.6	**66**/53
			10.5	1.44	3.42	7.7	2.4/3.7	180/100
*poly*(tpy)	THF Toluene	10.8	6.0	1.60	9.63	4.5	**2.8**/3.3	**45**/36
			6.2 *	1.58 *	6.63 *	5.5	**2.6**/3.3	**63**/44
(Fc)-*co*-(tpy)	THF	13.3	10.4	1.76	7.06	6.1	**2.7**/3.4	**81**/57
(Fc)-*bl*-(tpy)	THF	18.7	17.7	1.27	4.7	9.1	**2.9**/5.0	**250**/110

* The presented data were obtained for the independently synthesized sample.

**Table 4 polymers-14-01776-t004:** Summary of the conformational parameters evaluated for the studied macromolecules.

Sample	*M*_0_, g/mol	*M_L_*10^−9^, g/(mol cm)	*A_app_*, nm	*d*, nm	*L*/*A_app_*, cm^3^/g
*poly*(Fc)	368	14.6	**3.5**	1.38	13
*poly*(tpy)	416	16.5	**5.4**	1.62	6
(Fc)-*co*-(tpy)	395	15.7	**4.6**	1.47	11
(Fc)-*bl*-(tpy)	378	15.0	**4.3**	1.50	37
PMMA	100	3.97	**1.4** ^1^	0.81	

^1^ The obtained value represents the equilibrium rigidity, *A*.

## Data Availability

The data for the synthesis and structural characterization of all compounds are stored at the FSU Jena and St. Petersburg State University.

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
