# Peer review of "Hydrodynamic Characteristics and Conformational Parameters of Ferrocene-Terpyridine-Based Polymers"

_polymers, 2022, doi:10.3390/polym14091776_

Round 1

Reviewer 1 Report

In the current work, the authors investigated the hydrodynamic properties and conformational parameters of four polymers. The results are well represented & clearly interpreted. I highly recommend publishing the manuscript in Polymers. Here are minor comments which need to be considered by the authors:

  1. All acronyms need to be defined in the first time they appear in the manuscript. For example, RAFT, AIBN, THF, DMFA, CPDB, 1H-NMR,…etc. Also parameters in some equations are not defined such as those in equation 1 (kB, T). A list of abbreviation can be added.
  2. The used numbering scheme of figures in SI is not matching that given in the manuscript as two styles are used in the SI (For example Figure SI-1 , Figure S2, Figure S3). Thus, for consistency, one style to be used in SI and then edit them in the manuscript accordingly.
  3. The polymer codes in Figures 5-6 , and caption of Figures S4-S8 need to be revised as they are not matching with those given in Figure 1.
  4. The parameters in Table SI need to be defined in the Table caption/footnote. Also, the same need to be given in line 113-114.
  5. Line 61 & Line 69: please change (metalcontaining) to (metal-containing)
  6. Line 95: are presented instead of is presented
  7. Line 40: here the authors need to elaborate more in the synthetic difficulties.
  8. I suggest adding the structures of the monomers (1-2) in Figure 1 and instead of showing only the structure of the polymers it would be better to show the reaction scheme.
  9. Line 145 (footnote of Table 1): Change Mn to Mn
  10. Line 259: Change colons (: ) to a full stop (. )
  11. Line 253: please revise the caption of Figure 4 as symbol (a) is given in two positions.
  12. Line 340 & Line 344: The numbers 2.87x1023 &  52x10-8 to be written correctly.
  13. Line 378: the sentence needs to be revised as it is not clear!
  14. I suggest the authors to involve more recent references to the list of references.

Author Response

Dear Reviewer,

We are very grateful to you for your valuable comments and requests for the clarification. We performed thorough analysis of the manuscript and answered all the comments. In here presented data are the step-by-step answers (in blue font) to the reviewer’s comments. All the changes in the manuscript and supporting information files are highlighted with the yellow background.

_______

All acronyms need to be defined in the first time they appear in the manuscript. For example, RAFT, AIBN, THF, DMFA, CPDB, 1H-NMR,…etc. Also parameters in some equations are not defined such as those in equation 1 (kB, T). A list of abbreviation can be added.

>>>All acronyms and parameters are defined at the places in the manuscript, where they appeared for the first time. For example, k_B and T were introduced earlier in the text (page 6, at the end of ‘AUC’ section). The necessary corrections have been made in the manuscript.

The used numbering scheme of figures in SI is not matching that given in the manuscript as two styles are used in the SI (For example Figure SI-1 , Figure S2, Figure S3). Thus, for consistency, one style to be used in SI and then edit them in the manuscript accordingly. The polymer codes in Figures 5-6 , and caption of Figures S4-S8 need to be revised as they are not matching with those given in Figure 1. The parameters in Table SI need to be defined in the Table caption/footnote. Also, the same need to be given in line 113-114.

>>> The necessary corrections have been made in the manuscript.

Line 61 & Line 69: please change (metalcontaining) to (metal-containing)

Line 95: are presented instead of is presented

Line 40: here the authors need to elaborate more in the synthetic difficulties.

I suggest adding the structures of the monomers (1-2) in Figure 1 and instead of showing only the structure of the polymers it would be better to show the reaction scheme.

Line 145 (footnote of Table 1): Change Mn to Mn

Line 259: Change colons (: ) to a full stop (. )

Line 253: please revise the caption of Figure 4 as symbol (a) is given in two positions.

Line 340 & Line 344: The numbers 2.87x1023 & 52x10-8 to be written correctly.

Line 378: the sentence needs to be revised as it is not clear!

I suggest the authors to involve more recent references to the list of references.

>>> All of the above comments (Lines  61 to 378, together with sentence clarification and recent publications) have been taken care of.

Reviewer 2 Report

The manuscript of Gubarev et al. describes the synthesis of ferrocene/terpyridine polymers and their characterization with various analytical methods. The practical work has been performed competently, so is the scholarly presentation. The manuscript is well structured and easy to read. I cannot find any serious linguistic problems. The analysis of the experimental results was carefully carried out and the interpretations are sound. There are only very minor, more cosmetic comments:

#page 2, lines 76-83: I do not understand this paragraph. What does a Fe-Pt alloy have to do with the presented polymers? An alloy is a metallic substance, the presented (and envisaged) materials are coordination compounds. Please explain or/and re-write.

#please define the solvent DMFA. Dimethylformamide? Then please use the more common abbreviation "DMF". Please take care that many diagrams also use the abbreviation DMFA.

#section 2.2, synthesis: Whereas details on analytical measurements and examination of the experimental results are given, the procedure on polymer preparation is far from being complete. Practical all necessary information are missing which would make it very difficult to reproduce the synthesis. There are no weights, concentrations, volumes, molar relations, yields, etc. Please include this information, either in the manuscript or in the supporting information. I would also recommend writing an own paragraph devoted to the chemical characterization in section 3. ("Results").

#page 4, lines 140 and figure caption: "isopranol" -> "isopropanol"

#page 10, lines 326-327: "It is known that the presence of a large side radical in a polymer chain leads to an increase in the equilibrium rigidity [45]." What do the authors mean when using the term "large side radical"? Which radical? I can not identify any radicals in the polymer chain. What does this have to do with rigidity? Please explain.

In summary, the manuscript is clearly suitable for publication after addressing the comments indicated above. Again, I would like to point out that a more detailed description of the synthetic procedure is necessary. I cannot recommend an acceptance if this issue will not be addressed in the revised version.

Author Response

Dear Reviewer,

We are very grateful to you for your valuable comments and requests for the clarification. We performed thorough analysis of the manuscript and answered all the comments. In here presented data are the step-by-step answers (in blue font) to the reviewer’s comments. All the changes in the manuscript and supporting information files are highlighted with the yellow background.

___________

#page 2, lines 76-83: I do not understand this paragraph. What does a Fe-Pt alloy have to do with the presented polymers? An alloy is a metallic substance, the presented (and envisaged) materials are coordination compounds. Please explain or/and re-write.

>>> The paragraph has been rewritten.

#please define the solvent DMFA. Dimethylformamide? Then please use the more common abbreviation "DMF". Please take care that many diagrams also use the abbreviation DMFA.

>>> The used solvent is Dimethylformamide. The necessary corrections (change to DMF) have been made in the manuscript.

#section 2.2, synthesis: Whereas details on analytical measurements and examination of the experimental results are given, the procedure on polymer preparation is far from being complete. Practical all necessary information are missing which would make it very difficult to reproduce the synthesis. There are no weights, concentrations, volumes, molar relations, yields, etc. Please include this information, either in the manuscript or in the supporting information. I would also recommend writing an own paragraph devoted to the chemical characterization in section 3. ("Results").

>>> ‘Polymer synthesis’ paragraph have been reworked and Table SI-2. ‘Experimental details for the polymer synthesis.’ have been added to SI.

#page 4, lines 140 and figure caption: "isopranol" -> "isopropanol"

>>> The necessary corrections have been made in the manuscript.

#page 10, lines 326-327: "It is known that the presence of a large side radical in a polymer chain leads to an increase in the equilibrium rigidity [45]." What do the authors mean when using the term "large side radical"? Which radical? I can not identify any radicals in the polymer chain. What does this have to do with rigidity? Please explain.

>>> The wrong terminology has been used. Instead of "large side radical" the termin “side chain” was supposed to be used. The necessary corrections have been made in the manuscript. The bigger the side chain of a monomer, composing a linear polymer main chain, the greater the steric interaction of the side chains, which increase rotation hindrance of neighboring monomer units of a main polymer chain and results in increase of an equilibrium rigidity (Kuhn segment length). The sentence has been re-written. 

In summary, the manuscript is clearly suitable for publication after addressing the comments indicated above. Again, I would like to point out that a more detailed description of the synthetic procedure is necessary. I cannot recommend an acceptance if this issue will not be addressed in the revised version.

>>> The necessary corrections have been made in the manuscript.

Reviewer 3 Report

This is a really relevant paper, containing valuable and meritory result in various regards. The systems, namely, polymers which can interact with metallic ions via coordination centers impairing in them special, features which allow a variety of interaction. For such purpose, and from the point of view of basic polymer science, they present the novelty – and challenges – of their synthesis and physicochemical characterization in solution.

The ability of synthetizing four different polymer kinds, namely Fc and tpy homopolymers, their random copolymer and even their block copolymer is really impressive. The characterization by hydrodynamic techniques is really complete, as it includes intrinsic viscosity, sedimentation coefficient from analytical ultracentrifugation (AUC) and diffusion coefficient by dynamic light scattering (DLS), along with physicochemical properties of solvent and solute. Thus, in addition to the remarkable merit of the synthesis,  this paper is an exemplary piece of work illustrating the use of three classical hydrodynamic techniques with modern instrumentation, particularly fruitful in the characterization of these complex, speciality  polymers.

I therefore recommend publication of this paper. Along my reading I have noticed a few details of minor importance that the authors could consider in a revised, final version of the manuscript, which would ready for publication.

Page 4. I assume that the “RID-10A” in the SEC apparatus is a differential refractive index detector. I also assume that dispersity D in Table I is the Mw/Mn ratio.

Page 6 DLS measurements were carried out over a range of angles, theta, from 30 to 130 degrees, and therefore a range of q.  I don’t see how ethis dependence is presented and discussed.

Page 6. Eq. (2) The authors assume, as it usual in AUC that the density rho to be used (which determines the buoyant mass of the solute) should be the solution density, which is  well approximated by the solvent density rho_0. This is fully correct, but this ‘academic’ detail could be mentioned.

Page 7, Fig. 4 , and page 9. One may wonder whether the multiple pieks observed  c(s) distributions seen in Fig 4 are indicative of a multimodal molecular weight distribution, or simply artifacts of the SEDFiT analysis for a sample with wide but continuous polydispersity

Page 8. “… the thusly obtained ?â„Ž values were recalculated to diffusion coefficients  …” I think that the Rh values could be included in Table 3.

Page 11, Fig. 6. You may give please reference number in the list of references instead of year. Also, indicate that solid lines and open circles correspond (I think) to PMMA.

Author Response

Dear Reviewer,

We are very grateful to you for your valuable comments and requests for the clarification. We performed thorough analysis of the manuscript and answered all the comments. In here presented data are the step-by-step answers (in blue font) to the reviewer’s comments. All the changes in the manuscript and supporting information files are highlighted with the yellow background.

___________________

Page 4. I assume that the “RID-10A” in the SEC apparatus is a differential refractive index detector. I also assume that dispersity D in Table I is the Mw/Mn ratio.

>>> Yes, It is a differential refractive index detector. The necessary corrections have been made in the manuscript.

Page 6 DLS measurements were carried out over a range of angles, theta, from 30 to 130 degrees, and therefore a range of q. I don’t see how ethis dependence is presented and discussed.

>>> This dependences were introduced to SI (Table SI-3. The initial DLS data). Also, at page 6  the following was stated: “... Diffusion coefficients  of the studied macromolecules at finite concentration were determined from linear dependences  (1/tau)(q^2)…”, meaning that the observed dependences of (1/tau)(q^2) describe the diffusion processes.

Page 6. Eq. (2) The authors assume, as it usual in AUC that the density rho to be used (which determines the buoyant mass of the solute) should be the solution density, which is well approximated by the solvent density rho_0. This is fully correct, but this ‘academic’ detail could be mentioned.

>>> The necessary corrections have been made in the manuscript.

Page 7, Fig. 4 , and page 9. One may wonder whether the multiple pieks observed c(s) distributions seen in Fig 4 are indicative of a multimodal molecular weight distribution, or simply artifacts of the SEDFiT analysis for a sample with wide but continuous polydispersity.

>>> The observed multiple peaks in this particular case represent the samples with wide but continuous polydispersity. This analysis was made at page 8: ‘... the other polymers give several prominent peaks attributed to their higher dispersity.

Page 8. “… the thusly obtained ?â„Ž values were recalculated to diffusion coefficients …” I think that the Rh values could be included in Table 3.

>>> The necessary corrections have been made in the manuscript.

Page 11, Fig. 6. You may give please reference number in the list of references instead of year. Also, indicate that solid lines and open circles correspond (I think) to PMMA.

>>> The necessary corrections have been made in the manuscript.